# Four New Unusual Pentacyclic Triterpenoids from the Roots of *Jasminum sambac* (L.) Ait

**DOI:** 10.3390/molecules28135097

**Published:** 2023-06-29

**Authors:** Olagoke Zacchaeus Olatunde, Jianping Yong, Canzhong Lu

**Affiliations:** 1Fujian Institute of Research on the Structure of Matter, Chinese Academy of Sciences, Fuzhou 350002, China; olatunde@fjirsm.ac.cn; 2Xiamen Institute of Rare-Earth Materials, Chinese Academy of Sciences, Xiamen 361021, China; 3University of Chinese Academy of Sciences, Beijing 100049, China

**Keywords:** unusual pentacyclic triterpenoids, isolation, structural elucidation, NMR, XRD

## Abstract

Four new unusual pentacyclic triterpenoids (**1**–**4**) were isolated from the roots of *Jasminum sambac* (L.) Ait. Their structures were elucidated by 1D and 2D NMR analysis, single-crystal X-ray diffraction and HRESIMS.

## 1. Introduction and Structural Elucidation

*Jasminum* (Oleaceae) is a genus with over 200 species, which are native to Asia, Australia, Africa, and the southern Pacific Islands [1]. Phytochemical investigations on some of the Jasminum species have revealed the presence of secoiridoids, lignans, triterpenoids, flavonoids, and sesquiterpenoids [2,3,4,5,6,7,8,9]. The roots of *Jasminum sambac* (L.) Ait., which have anesthetic and analgesic effects, are used in traditional Chinese medicine for the treatment of insomnia, headaches, decayed teeth, and injuries from falls. It was recorded that the roots are thought to be one important ingredient of “Ma-Fei-San”, which was created by Tuo Hua and used for surgeries due to its significant anesthetic and analgesic effects. Consequently, it is essential to study the ingredients of the roots of *Jasminum sambac* (L.) Ait. in depth.

In our previous work, we isolated and confirmed some compounds (triterpenoid, sesquiterpenoids, lignans and glycoside) from the roots of *Jasminum sambac* (L.) Ait. [10,11]. Here, we report the isolation and elucidation of another four new unusual pentacyclic triterpenoids (**1**–**4**) (Their structures are shown in Figure 1) from the roots of *Jasminum sambac* (L.) Ait.

**Compound 1**, Golden yellow solid, M.p 61–62 °C, HPLC purity: 96.596%, retention time: 17.067 min. Crystal data: orthorhombic, space group P212121 (no. 19), a = 7.8148(3)Å, b = 13.1614(6)Å, c = 25.3786(11)Å, V = 2610.28(19)Å^3^, Z = 4, T = 272.00 K, μ(Mo Kα) = 0.091 mm^−1^, Dcalc = 1.284 g/cm^3^, 22,104 reflections measured (5.454° ≤ 2θ ≤ 54.36°), and 5766 unique (Rint = 0.0704, Rsigma = 0.0650) which were used in all calculations. The final R_1_ was 0.0536 (I > 2σ(I)) and wR_2_ was 0.1512.

The molecular formula was confirmed as “C_30_H_32_O_7_” through HRESIMS (*m*/*z*: found: 527.2040 [M + Na]^+^, calcd.: 527.2040). The ^1^H NMR spectrum showed six methyl groups at [*δ*_H_: 1.26 (d, *J* = 4.4 Hz, H-12′), 1.32 (d, *J* = 4.4 Hz, H-17), 1.45 (s, H-16), 1.72 (s, H-18), 2.24 (s, H-15), and 2.31 (s, H-11′)] ppm; two aromatic protons at [*δ*_H_: 6.84 (s, H-6) and 6.93 (s, H-6′)] ppm; five hydroxyl protons at [*δ*_H_: 5.35 (s, 4-OH), 8.99 (s, 4′-OH), 9.10 (s, 8-OH), 11.78 (s, 7-OH), and 12.53 (s, 5′-OH)] ppm. The ^13^C NMR together with the DEPT spectra revealed thirty signals, including: six methyls (C-15, C-16, C-17, C-18, C-11′, and C-12′); two methylenes (C-2 and C-3); two methines (C-10 and C-8′); three olefinic methines (C-6, C-2′, and C-6′), and seventeen quaternary carbons (C-1, C-4, C-5, C-7, C-8, C-9, C-11, C-12, C-13, C-14, C-1′, C-3′, C-4′, C-5′, C-7, C-9′ and C-10). The HMBC correlations (Figure 2): from 4-OH (*δ*_H_ 5.35) to C-16 (*δ*_C_ 28.9), C-3 (*δ*_C_ 34.8), and C-4 (*δ*_C_ 71.1); from 8-OH (*δ*_H_ 9.10) to C-6 (*δ*_C_ 124.0) and C-8 (*δ*_C_ 148.3); from 7-OH (*δ*_H_ 11.78) to C-13 (*δ*_C_ 116.1), C-7 (*δ*_C_ 143.4), and C-8 (*δ*_C_ 148.3); from 4′-OH (*δ*_H_ 8.99) to C-6′ (*δ*_C_ 123.9) and C-4′ (*δ*_C_ 148.4); and from 5′-OH (*δ*_H_ 12.53) to C-10′ (*δ*_C_ 114.8), C-5′ (*δ*_C_ 143.6), and C-4′ (*δ*_C_ 148.4) supported the “hydroxyl groups” at C-4, C-7, C-8, C-5′ and C-4′, respectively. ^1^H-^1^H COSY correlations (Figure 2): from H-6′/H-12′ and H-8′/H-2′/H-11′ coupled with the guidance of HMBC correlations: from H-2′ (*δ*_H_ 5.69) to C-8′ (*δ*_C_ 35.5) and C-10′ (*δ*_C_ 114.8); from H-6′ (*δ*_H_ 6.93) to C-9′ (*δ*_C_ 137.2), C-5′ (*δ*_C_ 143.5), and C-4′ (*δ*_C_ 148.4); from H-11′ (*δ*_H_ 2.31) to C-10′ (*δ*_C_ 114.8), C-7′ (*δ*_C_ 124.5), C-9′ (*δ*_C_ 137.2), and C-4′ (*δ*_C_ 148.4); and from H-12′ (*δ*_H_ 1.26) to C-8′ (*δ*_C_ 35.5) and C-9′ (*δ*_C_ 137.2) indicated the presence of the methylnaphthalen-1onyl group. Likewise, the correlations from H-10 (*δ*_H_ 4.01) to C-17 (*δ*_C_ 28.1), C-4 (*δ*_C_ 71.1), and C-12 (*δ*_C_ 137.1); from H-15 (*δ*_H_ 2.24) to C-13 (*δ*_C_ 116.1), C-5 (*δ*_C_ 123.8), and C-14 (*δ*_C_ 136.9); from H-17 (*δ*_H_ 1.32) to C-10 (*δ*_C_ 33.1) and C-14 (*δ*_C_ 136.9); from H-16 (*δ*_H_ 1.45) to C-3 (*δ*_C_ 34.8) and C-4 (*δ*_C_ 71.1), accompanied with ^1^H-^1^H COSY correlations of H-6/H-17 and H-10/H-15/H-17, revealed the existence of hydroanthracen-9-onyl moiety. The methylnaphthalen-1-onyl and hydroanthracen-9-onyl groups of **compound 1** were connected by analyzing the HMBC correlation from H-2′ (*δ*_H_ 5.69) to C-1 (*δ*_C_ 43.9), which was also supported by ^1^H-^1^H COSY correlation of H-2′/H-2/H-3. The ROESY correlations (Figure 3) showed that H-10 (*δ*_H_ 4.01) correlates with Me-16 (*δ*_H_ 1.45), Me-11′ (*δ*_H_ 2.31), and H-8′ (*δ*_H_ 4.17) correlates with Me-15 (*δ*_H_ 2.24), Me-18 (*δ*_H_ 1.72) and 4-OH (*δ*_H_ 5.35), which indicated its relative configuration. The detailed ^1^H and ^13^C NMR are shown in Table 1. The single-crystal X-ray diffraction analysis confidently confirmed its absolute configuration (CCDC 2259478; the XRD structure is shown in Figure 4).

**Compound 2**, Yellow solid, M.p 58–59 °C, HPLC purity: 92.29%, retention time: 18.807 min. The molecular formula was confirmed as “C_30_H_34_O_6_” through HRESIMS (*m*/*z*: 513.2247 [M + Na]^+^, calcd.: 513.2248). The ^1^H and ^13^C NMR data (Table 1) revealed six methyls at [*δ*_H_ 0.94 (d, *J* = 4.4 Hz, H-17), *δ*_H_ 0.99 (s, H-15), *δ*_H_ 1.26 (d, *J* = 4.5 Hz, H-12′), *δ*_H_ 1.51 (s, H-16), *δ*_H_ 1.58 (s, H-18), and *δ*_H_ 2.25 (s, H-11′)] ppm; one aromatic proton at [*δ*_H_ 6.89 (s, H-6′)] ppm; and three hydroxyl protons at [*δ*_H_ 5.40 (s, 4-OH), *δ*_H_ 9.07 (1H, s, 4′-OH) and *δ*_H_ 12.23 (s, 5′-OH)] ppm, which are identical to those of compound **1**. 

The ^1^H-^1^H COSY of H-17/H-5/H-6/H-7, H-15/H-10, H-16/H-15/H-3, H-18/H-11′/H-2, H-12′/H-6′/H-8 of compound **2** (Figure 2), guided by HMBC correlations (Figure 2) from H-17 (*δ*_H_ 0.94) to C-5 (*δ*_C_ 29.6) and C-7 (*δ*_C_ 42.4); from H-15 (*δ*_H_ 0.99) to C-6 (*δ*_C_ 38.3) and C-10 (*δ*_C_ 39.6); from H-12′ (*δ*_H_ 1.26) to C-8′ (*δ*_C_ 31.7) and C-9′ (*δ*_C_ 137.4); from H-16 (*δ*_H_ 1.51) to C-3 (*δ*_C_ 35.0) and C-4 (*δ*_C_ 70.1); from H-18 (*δ*_H_ 1.58) to C-2 (*δ*_C_ 33.1), C-1 (*δ*_C_ 43.8), C-11 (*δ*_C_ 135.5); and from H-11′ (*δ*_H_ 2.25) to C-8′ (*δ*_C_ 31.7), C-7′ (*δ*_C_ 123.6), C-9′ (*δ*_C_ 137.4) are aided in assigning the positions of methyl groups. In addition, the HMBC correlations from 4-OH (*δ*_H_ 5.40) to C-16 (*δ*_C_ 27.8), C-3 (*δ*_C_ 35.0) and C-4 (*δ*_C_ 70.1); from 4′-OH (*δ*_H_ 9.07) to C-6′ (*δ*_C_ 123.9), C-5′ (*δ*_C_ 143.5), and C-4′ (*δ*_C_ 148.7), and from 5′-OH (*δ*_H_ 12.23) to C-10′ (*δ*_C_ 115.8), C-5′ (*δ*_C_ 143.5), and C-4′ (*δ*_C_ 148.7) indicated the positions of the hydroxyl groups. The correlations from H-6′ (*δ*_H_ 6.89) to C-11′ (*δ*_C_ 17.7), C-9′ (*δ*_C_ 137.4), C-5′ (*δ*_C_ 143.5), and C-4′ (*δ*_C_ 148.7) revealed the position of aromatic proton. Its relative configuration of H-8′/H-5/4-OH and H-10/Me-16/Me-17/Me-11′ is supported by ROSY correlations (Figure 3). 

**Compound 3**, Yellow solid, M.p 73–74 °C, HPLC purity: 96.67%, retention time: 12.407 min. The molecular formula was confirmed as “C_30_H_34_O_6_” through HRESIMS (*m*/*z*: 513.2247 [M + Na]^+^, calcd.: 513.2248). The ^1^H and ^13^C NMR data are closely similar to those of compound **2**, except that the positions of the olefinic bond between C-13 and C-14 of compound **2** shifted to C-6 and C-14 of **compound 3**, which revealed the appearance of two peaks at *δ*_C_ 126.4 and *δ*_C_ 158.1 ppm for C-6 and C-14, respectively. This was also supported on the basis of HMBC correlations from H-5 (*δ*_H_ 6.47) to C-10 (*δ*_C_ 39.6), C-14 (*δ*_C_ 158.1), and C-9 (*δ*_C_ 187.8). The position of Me-17 in **compound 3** was assigned with the guidance of HMBC correlation from H-17 (*δ*_H_ 0.84) to C-13 (*δ*_C_ 39.6) and C-7 (*δ*_C_ 41.7) (Figure 2). The relative configuration of **compound 3** was similar to compound **2**, which was determined by ROESY (Figure 3). The detailed ^1^H and ^13^C NMR are shown in Table 1.

**Compound 4**, Yellow solid, M.p 96–98 °C, HPLC purity: 98.83%, retention time: 21.400 min. The molecular formula was confirmed as “C_30_H_40_O_5_” through HRESIMS (*m*/*z*: 503.2771 [M + Na]^+^, calcd.: 503.2768). The ^1^H NMR data showed four tertiary methyls at [*δ*_H_ 0.86 (s, H-17), *δ*_H_ 1.49 (s, H-16), *δ*_H_ 1.44 (s, H-18) and *δ*_H_ 0.93 (s, H-12′)] ppm and two secondary methyls at [*δ*_H_ 0.97 (d, *J* = 4.6 Hz, H-15) and *δ*_H_ 0.99 (d, *J* = 2.9 Hz, H-11′)] ppm. The ^13^C NMR together with DEPT revealed 30 carbon signals, including: six methyls (C-15, C-16, C-17, C-18, C-11′, and C-12′); six methylenes (C-2, C-3, C-5, C-6, C-6′, and C-7′); two oxygenated methines (C-9 and C-3′); three olefinic methines (C-8, C-2′, and C-4′); five sp^2^ quaternary carbons (C-11, C-12, C-13, C-1′, and C-10′); and six sp^3^ quaternary carbons (C-1, C-4, C-14, C-9′, C-7 and C-5′). The ^1^H and ^13^C NMR data (Table 1) resembled those of (1S*,5S*,10aR*)-1-[(8′,8a′-dimethyl-4′-oxo-1′,4′,6′,7′,8′,8a′-hexahydronaphthalene-2′-yl]-4-hydroxy-1,4,5,10atetramethyl-1,2,3,4,5,6,7,9,10,10a-decahydroanthracen-9-one [4] (which was isolated from the J. sambac roots), except that the two carbonyl groups of (1S*,5S*,10aR*)-1-[(8′,8a′-dimethyl-4′-oxo-1′,4′,6′,7′,8′,8a′-hexahydronaphthalene-2′-yl]-4-hydroxy-1,4,5,10a tetramethyl-1,2,3,4,5,6,7,9,10,10a-decahydroanthracen-9-one were replaced with hydroxyl groups of compound **4**. The hydroxyl groups of compound **4** were assigned according to HMBC correlations (Figure 2): from H-9 (*δ*_H_ 4.17) to C-17 (*δ*_C_ 18.1), C-1 (*δ*_C_ 41.9), C-4 (*δ*_C_ 68.8), C-13 (*δ*_C_ 139.1); and from H-3′ (*δ*_H_ 4.51) to C-12′ (*δ*_C_ 18.4), C-8′ (*δ*_C_ 30.6), C-9′ (*δ*_C_ 41.6), C-10′ (*δ*_C_ 136.6), and C-4′ (*δ*_C_ 139.2), which are accompanied by ^1^H-^1^H COSY correlations of H-9/H-15/H-8 and H-3′/H-2′/H-11′/H-4′. In addition, the ^13^C and DEPT spectra indicated the presence of two carbonyl groups, unlike those of (1S*,5S*,10aR*)-1-[(8′,8a′-dimethyl-4′-oxo-1′,4′,6′,7′,8′,8a′-hexahydronaphthalene-2′-yl]-4-hydroxy-1,4,5,10a tetramethyl-1,2,3,4,5,6,7,9,10,10a-decahydroanthracen-9-one. The positions of methyl groups at C-10 and C-8′ were established by HMBC correlations from Me-15 (*δ*_H_ 0.97) to C-5 (*δ*_C_ 25.6), C-10 (*δ*_C_ 30.7), and C-14 (*δ*_C_ 42.8) and Me-11′ (*δ*_H_ 0.99) to C-7′ (*δ*_C_ 25.4), C-8′ (*δ*_C_ 30.6) and C-9′ (*δ*_C_ 41.6). The ROESY correlations of H-3′/Me-11′/Me-12′/Me-16/H-10 and H-9/Me-17/Me-18/H-8′ (Figure 3) established its relative configuration.

## 2. Conclusions

In conclusion, we have isolated and confirmed four new unusual pentacyclic triterpenoids from the roots of *Jasminum sambac* (L.) Ait (The detailed NMR, HRESIMS, and HPLC data were shown in Appendix A). This work has discovered new compounds from the roots of *Jasminum sambac* (L.) Ait. and has also enriched understanding of the types of triterpenoids.

## Figures and Tables

**Figure 1 molecules-28-05097-f001:**
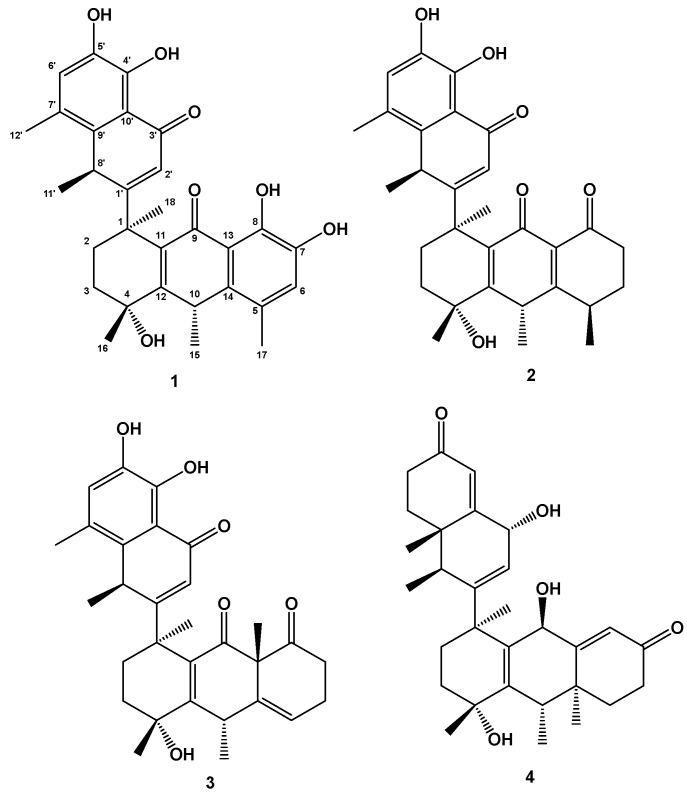
Structures of compounds (**1**–**4**).

**Figure 2 molecules-28-05097-f002:**
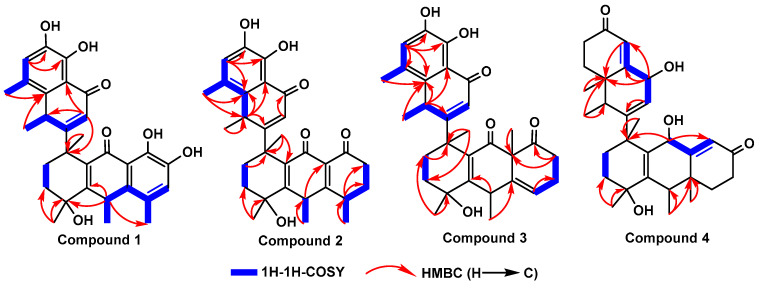
Structures with key ^1^H-^1^H COSY and HMBC correlations of the compounds **1** to **4**.

**Figure 3 molecules-28-05097-f003:**
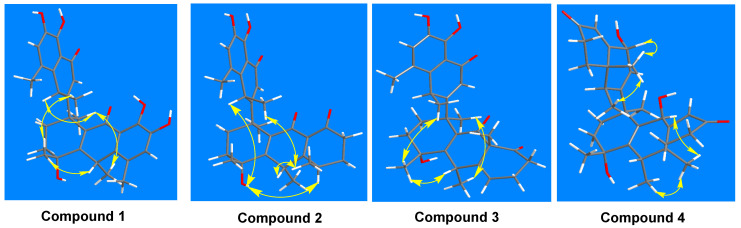
Key ROESY correlations of compounds **1** to **4**.

**Figure 4 molecules-28-05097-f004:**
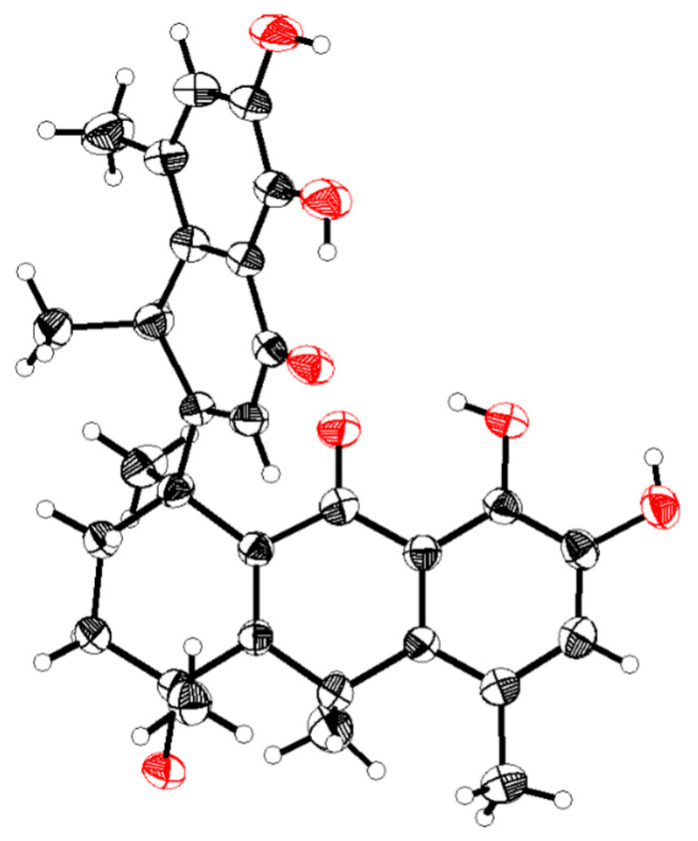
X-ray crystal structure for compound **1**.

**Table 1 molecules-28-05097-t001:** ^1^H(400 MHz) and ^13^C(100 MHz) NMR data for compounds **1** and **2** in DMSO-*d*_6._ and compounds **3** and **4** in CD_3_OD (*δ* in ppm and *J* in Hz).

No.	Compound 1		Compound 2		Compound 3		Compound 4	
	^1^H	^13^C	^1^H	^13^C	^1^H	^13^C	^1^H	^13^C
1		43.9 C		43.8		44.4 C		41.9 C
2α	1.98, m	39.2 CH_2_	1.90, m	33.1 CH_2_	1.85, dt(3.4, 2.5)	34.0 CH_2_	1.24, m	29.4 CH_2_
2β	2.12, m		1.53, m		2.18, m			
3α	1.80, m	34.8 CH_2_	1.61, m	35.0 CH_2_	2.02, m	35.5 CH_2_	1.76, m	33.7 CH_2_
3β	2.15, m							
4		71.1 C		70.1 C		71.3 C		68.8 C
5		123.8 C	1.22, m	29.2 CH_2_	6.47, s	126.4 CH	2.30, m	25.6 CH_2_
6α	6.84	123.9 CH	2.56, m	38.3 CH_2_	2.58, d(11.8)	38.5 CH_2_	1.60, m	26.0 CH_2_
6β			2.46					
7α		143.4 C	2.43, d(1.8)	42.4 CH_2_	2.39, d(2.1)	41.7 CH_2_		188.7 C
7β			2.21, m		2.27, m			
8		148.3 C		199.6 C		200.4 C	7.01, s	139.9 CH
9		191.9 C		191.9 C		187.8 C	4.17, s	67.9 CH
10	4.01, q(4.4)	33.1 CH	3.26, m	39.6 CH	3.76, m	39.9 CH	2.24	30.7 CH
11		137.1 C		135.5 C		136.7 C		
12		148.34		148.8 C		142.9 C		
13		116.1 C		135.6 C		39.6 C		139.1 C
14		136.9 C		157.7 C		158.1 C		42.8 C
15	2.24, s	17.7 CH_3_	0.99, s	17.6 CH_3_	1.04, s	16.8 CH_3_	0.97, s	14.0 CH_3_
16	1.45, s	28.9 CH_3_	1.51, s	27.8 CH_3_	1.45, s	27.2 CH_3_	1.49, s	26.3 CH_3_
17	1.32, d(4.4)	28.1 CH_3_	0.94, d(4.4)	15.3 CH_3_	0.84, d(4.8)	13.7 CH_3_	0.86, s	18.1 CH_3_
18	1.72, s	23.2 CH_3_	1.58, s	25.6 CH_3_	1.73, s	23.4 CH_3_	1.44, s	23.1 CH_3_
1′		168.9 C		167.5 C		168.8 C		138.5 C
2′	5.69, s	121.9 CH	6.27, d(0.7)	126.8 CH	6.89, s	122.9 CH	5.79, s	125.7 CH
3′		191.8 C		186.9 C		191.2 C	4.51, s	66.2 CH
4′		148.4 C		148.7 C		147.9 C	6.96, s	139.2 CH
5′		143.6 C		143.5 C		142.9 C		187.2 C
6′	6.93, s	124.0 CH	6.89, s	123.9 CH	6.90, s	122.4 CH	2.28, m	25.9 CH_2_
7′		124.5 C		123.6 C		123.7 C	1.58, m	25.4 CH_2_
8′	4.17, q(4.8)	35.5 CH	4.29, q(4.5)	31.7 CH	4.09, q(4.4)	33.1 CH	2.30	30.6 CH
9′		137.2 C		137.4 C		136.8 C		41.6 C
10′		114.8 C		115.8 C		115.2 C		136.6 C
11′	2.31, s	18.1 CH_3_	2.25, s	17.7 CH_3_	2.32, s	16.3 CH_3_	0.99, d(2.9)	13.9 CH_3_
12′	1.26, d(4.4)	24.4 CH_3_	1.26, d(4.5)	27.9 CH_3_	1.42, d(4.4)	26.7 CH_3_	0.93, s	18.4 CH_3_
4-OH	5.35, s		5.40, s					
7-OH	11.78,							
8-OH	9.10, s							
4′-OH	8.99, s		9.07, s					
5′-OH	12.53, s		12.23, s					

## Data Availability

Not applicable.

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
