# Peer review of "Four New Unusual Pentacyclic Triterpenoids from the Roots of Jasminum sambac (L.) Ait"

_molecules, 2023, doi:10.3390/molecules28135097_

Round 1

Reviewer 1 Report

This manuscript "Four new unusual pentacyclic triterpenoids from the roots of Jasminum sambac (L.) Ait" presents an interesting study on the isolation and elucidation of four new unusual pentacyclic triterpenoids from the roots of Jasminum sambac (L.) Ait. The authors have provided detailed information on the structure elucidation of the compounds, including 1H and 13C NMR data, HMBC and ROESY correlations.
Overall, this is an interesting study that provides valuable information on the isolation and elucidation of four new unusual pentacyclic triterpenoids from the roots of Jasminum sambac (L.) Ait. However, the manuscript requires major revisions before it can be accepted for publication.
The manuscript must include detailed information in the introduction, discussion, conclusions, and abstract. For instance, the authors should supply further information about the biological activity of the compounds, the steps taken in their isolation and purification, and the traditional Chinese medicine "Ma-Fei-San" and its components in the introduction.

Author Response

Dear professor,

   We are very thankful for your careful revision our manuscript, and provide constructive comments for us. Accordind to your comments, we have corrected our manuscript accordingly, and the corrected places were marked “ in red” in the manuscript. You can find them easily.

   As for the biological evaluation, the time is not enough for this work, for the first author will graduate so soon. In addition, the main work is to discover the new compounds from this plant medicine.

   Thanks a lot again for your careful revision our manuscript.

   Best regards,

   Jianping Yong & Canzhong Lu

Reviewer 2 Report

In this manuscript the authors report the structure elucidation of four new pentacyclic triterpenoids isolated from the roots of Jasminus sambac.

The manuscript is not appropriately organized as the basic Sections (Introduction, Results, Dicussion, Conclusion etc) are missing and the spectroscopic data are not discussed in detail, whereas, some critical aspects must be more thoroughly tackled, such for example the stereochemistry of compound 1, the only one they performed the X-ray crystallography on.

They claim that the crystallographic data together with Roesy spectrum analysis allow to assign the absolute stereochemistry at the stereocenters of the molecule. In my opinion, without an internal reference (for example the presence of a chiral centre with a known absolute configuration) that is quite hard to achieve.

Furthermore, the crystal structure reported in Fig 3 seems to be the enantiomer of the compound 1 depicted in Fig.1. In this regard, also in consideration that  in CCDC platform the crystal coordinates have not be found, the authors must properly clarify this issue.

Other minor concerns deal with proton resonance assignment; the authors have to more carefully checked their data, indeed, in some cases there are incoherence between the proton resonances in the spectra and the tabulated signals values (es in compound 3 the singlet assigned to CH3-17 is not centred at 0.84; furthermore, why in compound 4 the methyl-15 resonates as doublet instead of singlet (have they observed rotamers, in some ways?).

Finally, the authors must also check the appropriate name for the chemical structure, for example “methylnaphthalene-1-only……sounds incorrect.

English language requires minor editing

Author Response

Dear professor,

   We are very thankful for your careful revision our manuscript, and provide constructive comments for us. Accordind to your comments, we have corrected our manuscript accordingly, and the corrected places were marked “ in red” in the manuscript. You can find them easily.

This manuscript will be published as a communication, so the manuscript was

prepared not like the article.

    We have checked other points metioned by you and corrected them accordingly. All corrected places were marked “ in red “ and you can find them easily.

     Thanks a lot again for your careful revision our manuscript.

     Best regards,

    Jianping Yong & Canzhong Lu

Round 2

Reviewer 1 Report

The manuscript has been sufficiently improved to warrant publication in Molecules